# Mulan: A Multi-Level Alignment Model for Video Question Answering

**Yu Fu**[1,2], **Cong Cao**[1,2*], **Yuling Yang**[1,2], **Yuhai Lu**[3],
**Fangfang Yuan**[1,2*], **Dakui Wang**[1,2], **Yanbing Liu**[1,2]

[1]Institute of Information Engineering, Chinese Academy of Sciences, Beijing, China
[2]School of Cyber Security, University of Chinese Academy of Sciences, Beijing, China
[3]China Academy of Information and Communications Technology, Beijing, China
{fuyu1998,caocong,yangyuling,yuanfangfang,wangdakui,liuyanbing}@iie.ac.cn
luyuhai@caict.ac.cn

## Abstract

Video Question Answering (VideoQA) aims to answer questions about the visual content of a video. Current methods mainly focus on improving joint representations of video and text. However, these methods pay little attention to the fine-grained semantic interaction between video and text. In this paper, we propose Mulan: a Multi-Level Alignment Model for Video Question Answering, which establishes alignment between visual and textual modalities at the object-level, frame-level, and video-level. Specifically, for object-level alignment, we propose a mask-guided visual feature encoding method and a visual-guided text description method to learn fine-grained spatial information. For frame-level alignment, we introduce the use of visual features from individual frames, combined with a caption generator, to learn overall spatial information within the scene. For video-level alignment, we propose an expandable ordinal prompt for textual descriptions, combined with visual features, to learn temporal information. Experimental results show that our method outperforms the state-of-the-art methods, even when utilizing the smallest amount of extra visual-language pre-training data and a reduced number of trainable parameters. Our code is publicly available at https://github.com/fuyu1998/Mulan.

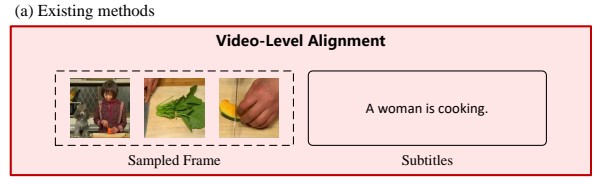

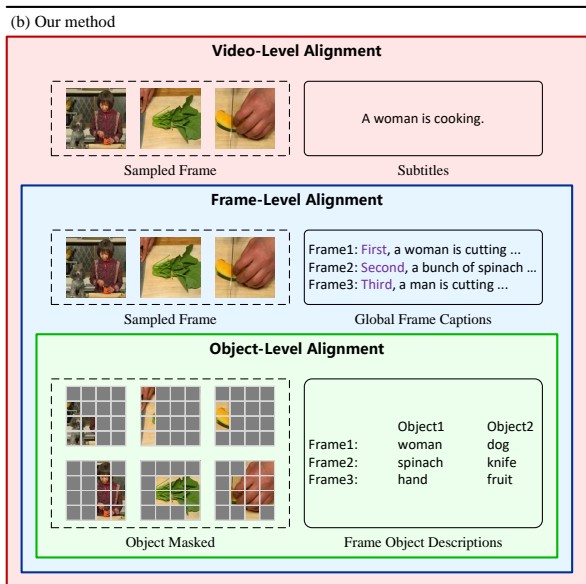

Figure 1: The difference between existing methods and our method. In contrast to existing methods that solely establish visual-language alignment at the video level, our method incorporates visual-language alignment at multiple levels.

## 1 Introduction

Video Question Answering (VideoQA) is a task that focuses on answering questions related to the visual content of a video. It serves as a representative task that showcases the fusion of visual and linguistic modalities. It demands the ability to comprehend and integrate data from both modalities to learn complex spatiotemporal information.

Given the videos are a continuous sequence of images, the mainstream models (Li et al., 2020, 2022c; Yang et al., 2021, 2022a; Zellers et al., 2021;

Xu et al., 2023) commonly adopt a two-step approach to establish a joint representation of video and text. First, an image-language model (He et al., 2016; Radford et al., 2021) is utilized to extract visual and textual features from the video frames. Then, alignment between the visual and textual modalities is established based on the extracted visual and textual features. However, existing methods align video and text globally to learn semantic correlations, disregarding the fine-grained interaction between local salient information in the video and important textual descriptions. Furthermore,

---
*Corresponding authors.

the majority of subtitles in the dataset are brief, resulting in a lack of detailed textual descriptions for significant video content, which further undermines the effectiveness of semantic alignment. Therefore, relying solely on global alignment is insufficient to address the semantic gap issue.

To address the above issues, we propose Mulan: a Multi-Level Alignment Model for Video Question Answering. Specifically, we establish alignment between visual and textual modalities at the **video-level**, **frame-level**, and **object-level**, as shown in Figure 1. Our method can be divided into three stages:

(1) Stage one involves the process of multi-level visual features generation. The image encoding of all sampled frames is regarded as video-level visual features. The image encoding of an individual frame is regarded as frame-level visual features. For object-level visual features, we have introduced a mask-guided object-level visual feature encoding method.

(2) Stage two involves the process of multi-level textual description generation. The original description of the video is regarded as video-level textual description. For frame-level textual descriptions, we have introduced the use of a caption generator. For object-level textual descriptions, we have proposed a vision-guided generative approach.

(3) Stage three involves the process of multi-level alignment and training. Specifically, we initially convert visual features into inputs for a language model and design prompts to concurrently establish alignment at the video-level, frame-level, and object-level. Finally, we employ a masked language modeling objective for training the language model, enabling it to acclimatize to visual input and establish multi-level alignment.

Our contributions can be summarised as follows:

(1) We introduce a multi-level visual-language alignment method for video question answering. To the best of our knowledge, this is the first work in the field of video question answering that explores multi-level visual-language alignment.

(2) We give a clear and unified spatiotemporal information learning framework. At the object-level and the frame-level, the method can learn spatial information, while acquiring temporal information at video-level.

(3) The experimental results illustrate that our method surpasses state-of-the-art baselines in video question answering tasks, even when using a mini-

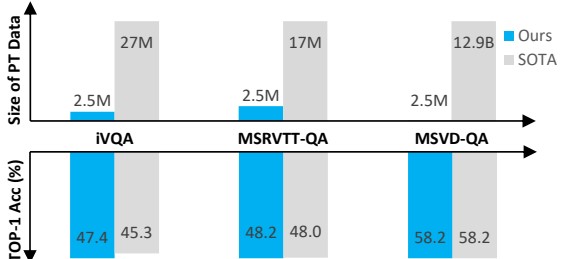

Figure 2: Comparison of the size of extra pre-training dataset and performance between our method and state-of-the-art methods. Our method demonstrates superior performance compared to state-of-the-art methods, even when utilizing extra pre-training datasets that are at least one order of magnitude smaller.

mal amount of visual-language extra pre-training data (Figure 2), and a reduced number of trainable parameters. Moreover, as the size of the pre-training dataset increases, the performance of our method can be further improved.

## 2 Related work

### 2.1 Pre-Trained Image-Language Models

In the realm of visual and language fusion methodologies, exploratory methods, as exemplified by the multi-stream approach(Lei et al., 2018), entail the utilization of a unified encoder to merge various information streams. This method facilitates the utilization of visual and language information at diverse levels of granularity and offers straightforward scalability. However, information of the same granularity is merged into distinct streams before alignment, resulting in a relatively poor semantic consistency among these streams.

In recent years, contrastive learning has been substantiated as an efficacious approach for acquiring cross-modal joint representations. The image-language pre-trained model, based on contrastive learning, aims to establish a mapping between global image features and global text features in a shared space (Radford et al., 2021; Li et al., 2022c; Lin et al., 2022; Jia et al., 2021; Wang et al., 2022b). These model considers aligned image-text pairs as positive samples and unaligned pairs as negative samples to optimize visual-language alignment.

Due to the requirement of a substantial amount of training data for video-language models to perform well, recent research has focused on transferring image-language models to video-text tasks (Yang et al., 2022a; Li et al., 2023b; Yeh et al., 2023). This is because there is a strong correlation

between images and videos. Sparse sampling offers an effective approach to video representation (Li et al., 2022a; Lei et al., 2021), contrasting with the use of 3D dense features (Feichtenhofer et al., 2019). It facilitates the more efficient utilization of pre-trained image-language models within the realm of video processing tasks (Wang et al., 2022e; Fang et al., 2021; Luo et al., 2022).

Our method continues along this line of methodology by ingeniously constructing a bridge from image-language to video-language through meticulous design, thereby facilitating the transfer of knowledge.

## 2.2 Language Models for Visual-Language Alignment

Some works focus on converting visual information entirely into symbolized textual information (Yang et al., 2022b; Wang et al., 2022e; Lin et al., 2023). Some other works use raw textual descriptions of videos, adopt the approach of freezing the weights of pre-trained language models, and integrate visual information to address tasks that involve both visual and language processing (Alayrac et al., 2022; Eichenberg et al., 2022; Yang et al., 2022a). However, the first category of methods disregards visual information, particularly fine-grained visual details. Conversely, the second category of methods lacks detailed textual descriptions and only achieves alignment at the video level.

In contrast, our method considers multi-level visual information and goes beyond by constructing semantic representations with multi-level visual-language alignment. In addition, we use lightweight Adapter layers (Houlsby et al., 2019; Hu et al., 2022) and frozen the language model. Our model can be easily applied to different types of language models, thereby facilitating practical application and deployment.

## 3 Method

### 3.1 Overall Framework

Our method can be divided into three stages, as illustrated in Figure 3: (1) multi-level visual feature generation, (2) multi-level textual description generation, (3) multi-level alignment and training.

Firstly, for a given sampled frame, an image encoder is applied to extract global frame visual features. Then, a mask generator is utilized to create masks for the objects within the frame, and the same image encoder is applied to obtain local frame visual features. Global frame visual features provide a comprehensive representation of the overall spatial information, while local frame visual features accurately capture specific object details such as position and attributes. It is worth noting that in this process, a mask generator is employed instead of an object detector. This choice is motivated by the fact that the output bounding boxes from an object detector are often not specific enough and can also struggle to accurately recognize objects in an open-vocabulary setting. Furthermore, due to the fact that both local frame visual features and global frame visual features are derived via a unified image encoder, it allows for the coexistence of local frame visual features and global frame visual features within the same semantic space.

Next, for global frame visual features, frame captions are generated using the image-language model BLIP (Li et al., 2022b). From these frame captions, the part-of-speech tagging tool provided by spaCy (Honnibal and Johnson, 2015) is employed to extract noun phrases, thus establishing a dynamic vocabulary tailored to the frame. The dynamic vocabulary terms are subsequently amalgamated with predefined static vocabulary, thereby creating the vocabulary input for the object filter. The object filter is then utilized to extract global and local frame objects from the respective global and local frame visual features. It is worth noting that in this process, due to the differences in the information contained in global and local frame visual features, the extracted global and local frame objects may also differ.

Finally, the global and local frame visual features, in conjunction with global and local frame object descriptions and global frame captions, serve as the input to a language model equipped with the adapter. Within the information fusion process, alignment between frame-level and object-level has been established. Simultaneously, temporal information is integrated by implementing an expandable ordinal prompt approach, thereby enabling the establishment of video-level alignment.

### 3.2 Multi-Level Visual Features Generation

To capture multi-level visual information, it is crucial to understand both the relationships between objects within video frames, which are represented as global frame visual features, and the local semantic information of the objects, which are represented as local frame visual features.

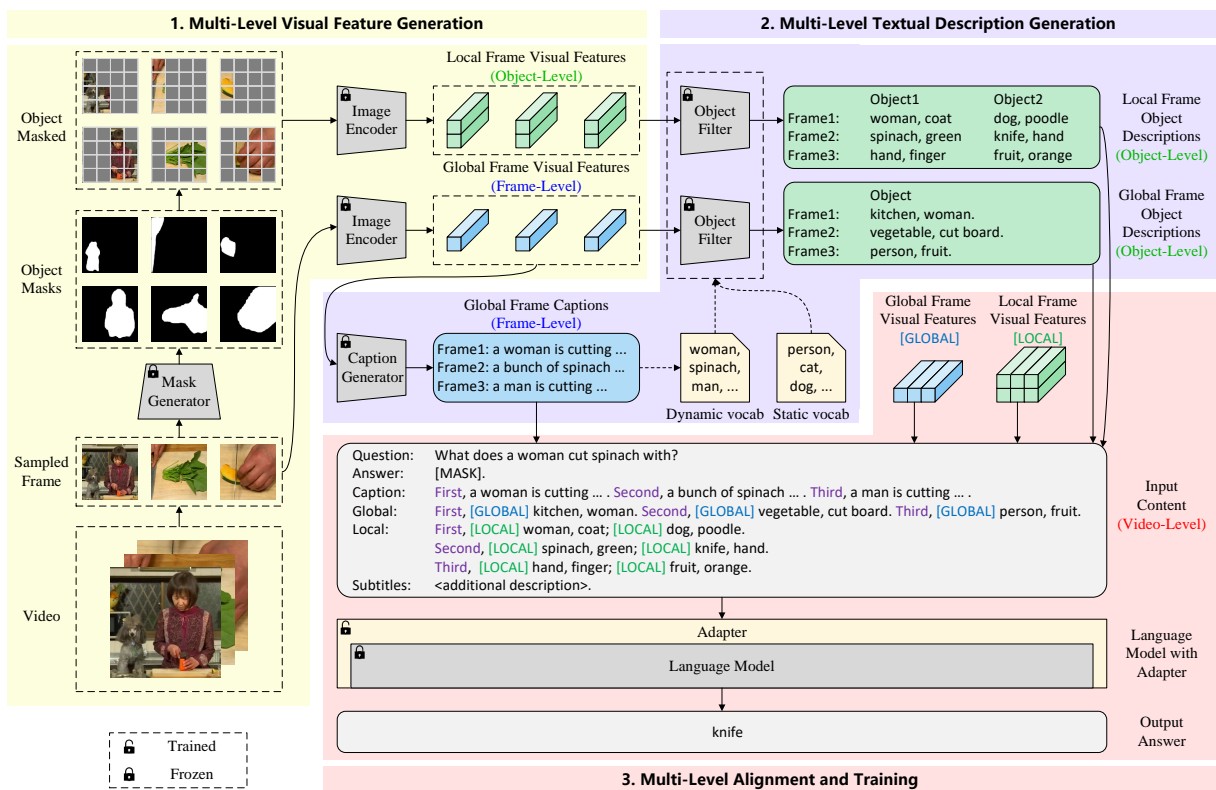

Figure 3: Overall framework of our method. (1) **Multi-Level Visual Feature Generation.** For each sampled frame, we use an image encoder to generate global frame visual features and utilize the mask generator to guide the generation of local frame visual features. (2) **Multi-Level Textual Description Generation.** We utilize a caption generator to extract frame captions and construct both dynamic and static vocabulary. The object filter is then utilized to generate frame objects based on these vocabularies. (3) **Multi-Level Alignment and Training.** We integrate visual and textual information to establish object-level and frame-level alignment, incorporating temporal cues to extend the alignment to the video level. Subsequently, we utilize a language model equipped with adapters to obtain the answers.

**Global Frame Visual Features.** The video is represented as a sequence of frames $f = \{f_i\}_1^T$ obtained through uniform and sparse sampling, where $T$ is the total number of sampled frames. Each frame $f_i$ is individually encoded using the image encoder $\phi_{\text{CLIP}}$ to generate global frame visual features $v^G$:

$$v_i^G = \phi_{\text{CLIP}}(f_i) \in \mathbb{R}^{D_v} \tag{1}$$

$$v^G = \{v_i^G\}_1^T \in \mathbb{R}^{T \times D_v} \tag{2}$$

where $D_v$ is the dimension of the visual features.

As $v^G$ represents the visual features of the original frames, it contains the overall spatial information of those frames. Therefore, we consider $v^G$ to represent frame-level visual information. During the experimental process, the image encoder is frozen, meaning that its weights are not updated.

**Local Frame Visual Features.** Our goal is to establish multi-level visual-language alignment.

Given the effectiveness of contrastive learning-based image-language models in learning shared representations of images and text in a common space, we employ CLIP (Radford et al., 2021), a contrastive learning-based model, as the image encoder. However, due to its focus on capturing global information of images, CLIP is not well-suited for directly capturing local details. To address this issue, we leverage the existing unsupervised mask generator CutLER (Wang et al., 2023b) to guide the image encoder CLIP in generating local frame visual features.

To obtain local frame visual features, we first utilize the mask generator CutLER $\phi_{\text{CutLER}}$ to generate a set of image masks $m_i$ for each frame $f_i$. We refer to the collection of mask sets for each frame as $m$:

$$m_i = \phi_{\text{CutLER}}(f_i) \tag{3}$$

$$m = \{m_i\}_1^T \tag{4}$$

Then, we apply cropping and masking operations to the images and feed them into the image encoder $\phi_{\text{CLIP}}$ to obtain the local frame visual features $v^L$:

$$v_i^L = \phi_{\text{CLIP}}(\mathcal{T}_{crop}(f_i \odot m_i)) \in \mathbb{R}^{N_i \times D_v} \quad (5)$$

$$v^L = \{v_i^L\}_1^T \in \mathbb{R}^{(\sum_{i=1}^T N_i) \times D_v} \quad (6)$$

where $\mathcal{T}_{crop}(\cdot)$ denotes the operations of cropping and masking, and $\odot$ is the Hadamard product operation. We assume that the number of masks for the $i$-th frame is $N_i$.

Due to the removal of object-irrelevant regions in $v^L$, it focuses solely on the target objects themselves. Therefore, we consider $v^L$ to represent object-level visual information. During the experimental process, the mask generator is also frozen.

### 3.3 Multi-Level Textual Description Generation

To capture multi-level textual information, it is crucial to utilize both the text that describes the relationships between objects, represented by the global frame captions, and the detailed information about the objects, represented by the descriptions of global and local frame objects.

**Global Frame Captions.** To establish textual connections among objects in the frame and obtain an overall description of the frame's information, we utilize the image-language model BLIP (Li et al., 2022b) to generate global frame captions $c = \{c_i\}_1^T$. We consider $c$ to represent frame-level textual information.

**Global and Local Frame Object Descriptions.** Global frame captions alone may not capture the fine-grained details of objects. Therefore, constructing textual descriptions only at the frame level is insufficient. To address this issue, we further utilize global and local frame visual features to construct global and local object descriptions.

Firstly, we construct a predefined static vocabulary $V^S$, which includes a set of candidate object names and attributes. However, the predefined static vocabulary cannot cover all objects and attributes. To address this issue, we utilize spaCy to extract noun phrases from the global frame captions $c$, creating a dynamic vocabulary $V^D$. The final vocabulary $V$ is defined as follows:

$$V = V^S \cup V^D \quad (7)$$

Next, we utilize the text encoder $\phi_{\text{CLIP-text}}$ in the CLIP model, which is based on contrastive

learning, to compute the text features $r$ for the final vocabulary $V$:

$$r = \{\phi_{\text{CLIP-text}}(V_i)\}_1^L \quad (8)$$

where $L$ is the size of the final vocabulary $V$.

Finally, we determine object descriptions for each visual feature by evaluating the cosine similarity between the visual features $v^G$ or $v^L$ and text features $r$. For the global frame visual features $v^G$, we generate a total of $M^G$ global object descriptions for each visual feature, denoted as $t^G$. Similarly, for the local frame visual features $v^L$, we generate a total of $M^L$ local object descriptions for each visual feature, denoted as $t^L$.

Due to the different focuses of global frame visual features $v^G$ and local frame visual features $v^L$, there are also differences in the global object descriptions $t^G$ and local object descriptions $t^L$. We consider $t^G$ and $t^L$ to represent object-level textual information.

### 3.4 Multi-Level Alignment and Training

**Multi-Level Alignment Prompting.** To achieve Visual-Language alignment, we integrate the obtained object-level and frame-level visual features with their corresponding textual descriptions. This integration allows us to align and fuse the visual and textual information at both the object-level and frame-level. Then, we integrate expandable temporal markers to facilitate visual-language alignment at the video-level.

As illustrated in Figure 3, we have devised the following prompt:

"Question: <Question>? Answer: [MASK]. Caption: <Caption>. Global: <Global Objects>. Local: <Local Objects>. Subtitles: <additional description>."

For the alignment at the **object-level**, we consider "a visual feature and its corresponding object description" as the alignment unit. We iteratively combine these units to align the global and local frame visual features ($v^G$ or $v^L$) with their respective object descriptions ($t^G$ or $t^L$). To leverage the inductive bias of language proximity, we sort the object descriptions in descending order based on their cosine similarity between text features and visual features. Specifically, for each global frame visual feature "[GLOBAL]" and local frame visual feature "[LOCAL]", we construct the following prompt:

"[GLOBAL] <Description 1>, ..."

"[LOCAL] <Description 1>, ..."

For the alignment at the **frame-level**, we consider all object-level information grouping by frame and global frame captions "First, <Caption 1>. Second, <Caption 2>. ..." as the alignment unit.

For the alignment at the **video-level**, we consider the frame-level information and additional descriptions as the alignment unit. At the video-level, temporal information is indeed crucial for video understanding. Therefore, we incorporate expandable ordinal markers into our model, such as "First," "Second," and so on.

**Language Model with Adapter.** We use the same Adapter layer as in the FrozenBiLM (Yang et al., 2022a) model. Each visual feature is passed through a linear projection layer and incorporated into the language model as an individual prompt. The global and local visual features ($v^G$ and $v^L$) are linearly mapped using the projection matrices ($P^G$ and $P^L$) to obtain the global and local feature prompts ($u^G$ and $u^L$):

$$u^G = \{P^G(v_i^G)\}_1^T \qquad (9)$$

$$u^L = \{P^L(v_i^L)\}_1^{\sum_{j=1}^T N_j} \qquad (10)$$

For autoencoder language models like De-BERTa (He et al., 2021), we employ a frozen MLM classifier head, denoted as $m^\theta$, to predict answers from the vocabulary set $A$, which is constructed from the answers appearing in the training set.

**Multimodal Training.** During training, we only update the parameters of the visual-to-text projection modules $P$ and the adapter module. To ensure consistency with language models, We use the visually-conditioned Masked Language Modeling (MLM) objective, where some tokens $\{x_m\}$ are randomly masked, and the model has to predict these masked tokens based on the other tokens and the visual input. Formally, we minimize the following loss:

$$\mathcal{L}_\mu(x, y) = -\frac{1}{M} \sum_m \log p_\mu(\tilde{x}, y)_m^{x_m}, \qquad (11)$$

where $\tilde{x}$ is the corrupted text sequence, $y$ is the sequence of visual input, $p_\mu(\tilde{x}, y)_m^{x_m}$ is the probability for the (masked) $m$-th token in $\tilde{x}$ to be $x_m$, and $M$ is the number of masks in the sequence $\tilde{x}$. In detail, we adopt the same configuration as BERT (Devlin et al., 2019) for our method.

## 4 Experiments

### 4.1 Experimental Setup and Datasets

**General Setup.** To ensure reproducibility, we utilize the same fixed seed for all experiments. Each experiment was conducted three times, and the average results were reported as the final outcome. For all other methods, we report the experimental results as documented in the paper.

**Pre-Trained Models.** We utilize the CLIP ViT-L/14 model (Radford et al., 2021) as the image encoder $\phi_{\text{CLIP}}$ with an input image resolution of $336 \times 336$. We utilize the Cutler (Wang et al., 2023b) model as the mask generator $\phi_{\text{CutLER}}$, which leverages the Cascade Mask R-CNN (Cai and Vasconcelos, 2021) as the detector for generating masks. We utilize the BLIP model (Li et al., 2022b) with ViT-L (Dosovitskiy et al., 2021) as the caption generator. We utilize the DeBERTa-V2-XLarge model (He et al., 2021) as the language model.

Note that in practical applications, the caption generator usually incorporates an image encoder (Li et al., 2022b, 2023a; Wang et al., 2022c), which facilitates the integration of the image encoder and caption generator to streamline the process. In our experiments, we intentionally separated the image encoder and subtitle generator to ensure a fairer comparison with other CLIP-based methods.

**Datasets.** We pre-trained our model using the WebVid-2M (Bain et al., 2021) dataset. We conducted evaluations on the iVQA (Yang et al., 2021), MSRVTT-QA (Xu et al., 2017), MSVD-QA (Xu et al., 2017), and TGIF-FrameQA (Jang et al., 2017) datasets to assess the performance of our approach. Detailed information about the dataset is provided in the **Appendix A**.

Furthermore, given that pre-trained models are trained with additional datasets before being used, our method also incorporates these additional datasets. Our method predominantly integrates 400 million image-text pairs from the image encoder CLIP, along with 14 million image-text pairs from BLIP(Li et al., 2022b), and an additional 1.3 million image-text pairs from CutLER(Wang et al., 2023b). Therefore, we incorporate a total of 415.3 million image-text pairs through all pre-trained models.

**Implementation Details.** Detailed implementation information is provided in the **Appendix B**.

| Method | Trained Params | Extra PT Data | iVQA | MSRVTT-QA | MSVD-QA | TGIF-FrameQA |
|---|---|---|---|---|---|---|
| FrozenBiLM (Yang et al., 2022a) | **30M** | 10M | 39.6 | 47.0 | 54.8 | 68.6 |
| HiTeA (Ye et al., 2022) | 297M | 17M | — | 45.9 | 55.3 | 73.2 |
| mPLUG-2 (Xu et al., 2023) | — | 17M | — | 48.0 | 58.1 | **75.4** |
| Flamingo-3B (Alayrac et al., 2022) | 1.4B | 27M | 37.7 | 25.6 | 42.6 | — |
| Flamingo-9B (Alayrac et al., 2022) | 1.8B | 27M | 40.7 | 29.4 | 47.2 | — |
| Flamingo (Alayrac et al., 2022) | 10B | 27M | 45.3 | 47.4 | 52,3 | — |
| LAVENDER (Li et al., 2023b) | 198M | 30M | — | 45.0 | 56.6 | 73.5 |
| Just Ask (Yang et al., 2021) | 157M | 69M | 35.4 | 41.8 | 47.5 | — |
| VideoCoCa (Yan et al., 2022) | 2.1B | 100M | 39.0 | 46.3 | 56.9 | — |
| InternVideo (Wang et al., 2022d) | 1.3B | 110M | — | 47.1 | 55.5 | 72.2 |
| MERLOT (Zellers et al., 2021) | 223M | 180M | — | 43.1 | — | 69.5 |
| All-in-one (Wang et al., 2023a) | 110M | 283M | — | 46.8 | 48.3 | 66.3 |
| GIT (Wang et al., 2022a) | 700M | 800M | — | 43.2 | 56.8 | 72.8 |
| GIT2 (Wang et al., 2022a) | 5.1B | 12.9B | — | 45.6 | **58.2** | 74.9 |
| Mulan | 31M | **2.5M** | **47.4** | **48.2** | **58.2** | 72.0 |

Table 1: Comparison with the state of the art on fully-supervised benchmarks. Top-1 accuracy is reported. "Extra PT Data" refers to the quantity of extra video-text pairs and image-text pairs utilized during the pre-training phase. All methods are sorted in ascending order based on the "Extra PT Data" from smallest to largest. Throughout the paper, we utilize **bold** and underline formatting to emphasize the top two results.

| Method | Trained Params | Extra PT Data | iVQA | MSRVTT-QA | MSVD-QA | TGIF-FrameQA |
|---|---|---|---|---|---|---|
| CLIP (Radford et al., 2021) | — | — | 9.2 | 2.1 | 7.2 | 3.6 |
| HiTeA (Ye et al., 2022) | 297M | 5M | — | 8.6 | 18.2 | — |
| HiTeA (Ye et al., 2022) | 297M | 5M | — | 21.7 | 37.4 | — |
| FrozenBiLM (Yang et al., 2022a) | **30M** | 10M | 26.9 | 16.7 | 33.8 | 41.9 |
| UnFrozenBiLM (Yang et al., 2022a) | 890M | 10M | 21.0 | 17.6 | 31.9 | 30.7 |
| mPLUG-2 (Xu et al., 2023) | — | 17M | — | **43.8** | **55.3** | — |
| Flamingo-3B (Alayrac et al., 2022) | 1.4B | 27M | 32.7 | 11.0 | 27.5 | — |
| Flamingo-9B (Alayrac et al., 2022) | 1.8B | 27M | 35.2 | 13.7 | 30.2 | — |
| Flamingo (Alayrac et al., 2022) | 10B | 27M | **40.7** | 17.4 | 35.6 | — |
| Just Ask (Yang et al., 2021) | 157M | 69M | 13.3 | 5.6 | 13.5 | — |
| Mulan | 31M | **2.5M** | 35.7 | 20.3 | 38.7 | **55.6** |

Table 2: Comparison with the state of the art on zero-shot benchmarks. Top-1 accuracy is reported. We gray out the method additionally supervised pre-training on VQA v2(Antol et al., 2015) dataset.

## 4.2 Comparison with State-of-the-art

In this section, we conduct a comprehensive evaluation of our method by comparing it with state-of-the-art methods in both the fully-supervised and zero-shot settings.

**Fully-Supervised Benchmarks.** Table 1 presents the results of our method compared to the state-of-the-art fully-supervised methods. Our method achieves state-of-the-art performance on the iVQA, MSRVTT-QA and MSVD-QA datasets, while obtaining competitive results on the TGIF-FrameQA dataset. Note that we have utilized the smallest extra pre-training dataset compared to all methods. In particular, our method outperforms FrozenBiLM (Yang et al., 2022a), which also utilizes a language model with Adapter. This can be attributed to the effective transfer of the zero-shot capability of the mask generator, image encoder, and caption generator in our method, which reduces

the dependency on additional pre-training data.

**Zero-Shot Benchmarks.** Table 2 presents the results of our method compared to the state-of-the-art zero-shot methods. The few-shot results of our method can be found in **Appendix C**. Our method achieves state-of-the-art performance on the TGIF-Frame dataset, while obtaining competitive results on the iVQA dataset. Similarly, we have utilized the smallest extra pre-training dataset and a reduced number of trained parameters. Similar to a fully-supervised setting, our method demonstrates a relative advantage over FrozenBiLM (Yang et al., 2022a) in the zero-shot setting. In zero-shot tasks, the size of the pre-training dataset plays a more crucial role compared to fully-supervised tasks. However, even with the smallest extra pre-training dataset, our method still achieves state-of-the-art or competitive performance. This can be attributed to the effective transfer of the zero-shot capability

|   | Video Level | Frame Level | Object Level | iVQA | MSRVTT-QA | MSVD-QA | TGIF-FrameQA |
|---|---|---|---|---|---|---|---|
| 1 | ✓ | ✗ | ✗ | 26.5 | 16.3 | 31.0 | 25.3 |
| 2 | ✓ | ✓ | ✗ | 35.5 | 19.6 | 38.3 | 36.7 |
| 3 | ✓ | ✗ | ✓ | 30.3 | 18.3 | 35.3 | 47.5 |
| 4 | ✓ | ✓ | ✓ | **35.7** | **20.3** | **38.7** | **55.6** |

Table 3: Ablation studies on frame-level and object-level alignment utilizing video-level alignment.

|   | Video Level | Frame Level | Object Level | iVQA | MSRVTT-QA | MSVD-QA | TGIF-FrameQA |
|---|---|---|---|---|---|---|---|
| 1 | ✗ | ✓ | ✗ | 24.6 | 9.7 | 18.8 | 42.7 |
| 2 | ✗ | ✗ | ✓ | 24.5 | 10.7 | 19.8 | 42.9 |
| 3 | ✗ | ✓ | ✓ | 27.5 | 11.3 | 21.2 | 47.5 |
| 4 | ✓ | ✓ | ✓ | **35.7** | **20.3** | **38.7** | **55.6** |

Table 4: Ablation studies on frame-level and object-level alignment without utilizing video-level alignment.

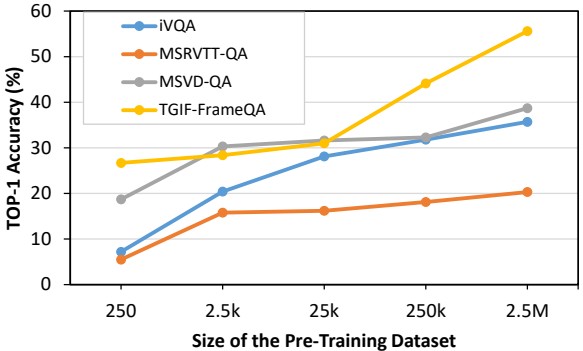

Figure 4: The influence of the size of the pre-training dataset on performance.

of the mask generator, image encoder, and caption generator in our method, which reduces the dependency on additional pre-training data.

The influence of the size of the pre-training dataset is shown in Figure 4, and detailed data can be found in Appendix C.2. For all datasets, our method exhibits enhanced performance as the size of the pre-training dataset expands. This indicates that our method still has significant potential for improvement.

**Dataset-Specific Analysis.** For the TGIF-FrameQA dataset, 51% of the questions focus on concrete object-related information such as color and quantity, while 49% of the questions address abstract information, such as actions. Our method performs well in capturing concrete object-related information in the zero-shot setting, resulting in a strong performance in that scenario. However, in the fully-supervised setting, our method faces challenges in further improving its capability to extract concrete information, and it exhibits relatively low ability in capturing abstract information

compared to concrete information. As a result, our method achieves suboptimal performance in the fully-supervised setting.

For the iVQA dataset, in the zero-shot setting, our method demonstrates comparable performance to Flamingo-9B, which has 1.8 billion trained parameters. In practice, it is commonly observed that larger trained parameter sizes tend to lead to improved generalization performance. Therefore, although our method achieved suboptimal performance compared to Flamingo, which has 10 billion trained parameters, it is still a significant accomplishment considering that our trained parameters constitute only 0.3% of Flamingo and 1.7% of Flamingo-9B.

### 4.3 The Influence of Multi-Level Alignment

In this section, we conduct an ablation study to analyze the influence of multi-level alignment and explore the complementary and substitutive effects among the three levels of alignment. All experiments were conducted with the identical pre-training configuration, as elucidated in **appendix B**. To minimize the influence of fine-tuning and conserve computational resources, all ablation experiments will be conducted and reported under the zero-shot setting.

**The Complementary Effect.** Table 3 presents the results of the ablation studies on frame-level and object-level alignment, with the utilization of video-level alignment. This experiment aims to uncover the complementary effects of frame-level and object-level alignment on video-level alignment.

For the MSRVTT-QA and MSVD-QA datasets, where the primary question types involve global descriptions of the videos, and for the iVQA dataset,

which focuses on scene and coarse-grained object information, frame-level alignment plays a dominant role as it encompasses a significant majority of the spatiotemporal information relevant to the questions. In contrast, for the TGIF-FrameQA dataset, which primarily focuses on object information, object-level alignment assumes the leading role as it helps capture fine-grained spatial information of the objects.

Overall, the comparison between row 4 and the other rows in Table 3 supports the complementary effect of object-level and frame-level alignment, and further confirms that utilizing three levels of alignment simultaneously is the optimal method.

**The Alternative Effect.** Table 4 presents the results of the ablation studies on frame-level and object-level alignment, without the utilization of video-level alignment. This experiment aims to uncover the alternative effects of frame-level and object-level alignment on video-level alignment.

Overall, the comparison between row 3 and rows 1 to 2 in Table 4 provides evidence that the combined effect of frame-level and object-level alignment is superior to their individual effects. Additionally, the comparison between row 4 and rows 1 to 3 confirms the importance of video-level alignment, suggesting that object-level and frame-level alignment have less substitutive effects on video-level alignment.

**Experimental Summary.** Considering all the conclusions, it can be inferred that the three levels of alignment exhibit complementary effects. Additionally, we investigate the influence of misalignment between the visual and textual modalities in **Appendix D**. For the influence of temporal information, please refer to **Appendix E**. Moreover, we conducted experiments to examine the influence of accumulated errors in the **Appendix F**.

## 5   Conclusions

In this paper, we introduce Mulan: A Multi-Level Alignment Model for Video Question Answering, which enhances temporal and spatial information learning through multi-level alignment. Additionally, we conduct experiments on public VideoQA datasets and achieve state-of-the-art results. Finally, we provide ablation studies to demonstrate the complementary effects of multi-level alignment and the effectiveness of our method.

## Limitations

In this work, we analyze the limitations as follows:

(1) Influence of maximum sequence length: Our method is influenced by the maximum sequence length of the language model. Although, as discussed in Appendix E, there is a diminishing marginal utility of increasing the number of sampled frames, and expanding sequence lengths is a crucial direction in the development of large language models. However, reducing the temporal and spatial redundant information between different levels can still be beneficial.

(2) Abstraction ability: Our method provides less assistance in answering questions that focus on abstract information, while it offers greater assistance in answering questions that focus on concrete information. To enhance the abstraction ability of the model, we believe that deeper interactions between different levels and modalities are beneficial.

## Ethics Statement

This work does not involve any direct ethical concerns. We are dedicated to advancing the field of video question answering. All experiments were conducted on open datasets, and the findings and conclusions of this paper are accurately and objectively reported.

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

## A    Datasets

**WebVid-2M** (Bain et al., 2021) consists of 2.5 million videos and their corresponding text pairs, collected from the Shutterstock website. The dataset does not include audio, and the video captions are obtained from pre-existing alternative textual descriptions.

**iVQA** (Yang et al., 2021) is an open-end VideoQA benchmark that focuses on objects, scenes, and people in instructional videos. It consists of 10k video clips and 10k QA pairs, split into 6k/2k/2k for training/validation/testing.

**MSVD-QA** (Xu et al., 2017) is an open-end VideoQA benchmark that consists of 2k video clips and 51k QA pairs, split into 32k/6k/13k for training/validation/testing. The QA pairs in MSVD-QA are automatically generated from video descriptions.

**MSRVTT-QA** (Xu et al., 2017) is an open-end VideoQA benchmark that consists of 10k video clips and 243k QA pairs, split into 158k/12k/73k for training/validation/testing. Similar to MSVD-QA, the QA pairs in MSRVTT-QA are also automatically generated from video descriptions.

**TGIF-FrameQA** (Jang et al., 2017) is an open-end VideoQA benchmark based on GIF videos. In the TGIF-FrameQA dataset, the majority of videos have short durations, typically less than 5 seconds. These videos are primarily classified into four categories: objects, quantities, colors, and positions. It consists of 46k gifs and 53k QA pairs, split into 39k/13k for training/testing.

## B    Implementation Details

**Hyperparameter Settings.** For all experiments, we employ uniform sampling to select 8 frames ($T = 8$) from each video. For pre-training experiments, we solely utilize video-level textual descriptions to avoid the model gathering information from other frame-level and object-level textual descriptions, which could potentially impact performance. For evaluation and fine-tuning experiments, we impose certain constraints to accommodate the token limitations of the language model. Specifically, we limit the generation of a maximum of 8 objects per frame. Additionally, the maximum number of global frame object descriptions per frame is set to 8, and the maximum number of local frame object descriptions per object is set to 2. For pre-training experiments, we use a sequence length of 128, and for evaluation and fine-tuning experiments, we use a sequence length of 512. The visual feature dimension of the pre-trained image encoder ViT-L/14 (Dosovitskiy et al., 2021) is 768, while the hidden dimension of the pre-trained language model DeBERTa-V2-XLarge (He et al., 2021) is 1536.

**Training.** We conducted all experiments on 8 NVIDIA GeForce RTX 3090 GPUs. The pre-training on the WebVid-2M (Bain et al., 2021) dataset took approximately 10 hours, with a total of 2 epochs. For the pre-training experiments, we set the learning rate to $3 \times 10^{-5}$ and trained for 2 epochs with a batch size of 16. For the fine-tuning experiments, we set the learning rate to $5 \times 10^{-5}$ and trained for 40 epochs with a batch size of 4. For all training experiments, we employ the Adam optimizer (Kingma and Ba, 2015) with $\beta = (0.9, 0.95)$ and no weight decay.

**Static Vocabulary Setting.** We constructed the static vocabulary based on the class names from the OpenImage v7 dataset (Kuznetsova et al., 2020),

| Supervision | iVQA | MSRVTT-QA | MSVD-QA | TGIF-FrameQA |
|---|---|---|---|---|
| 0% (zero-shot) | 35.7 | 20.3 | 38.7 | 55.6 |
| 1% (few-shot) | 41.3 | 37.6 | 48.7 | 63.7 |
| 10% (few-shot) | 43.0 | 42.9 | 52.3 | 67.7 |
| 100% (fully-supervised) | **47.4** | **48.2** | **58.2** | **72.0** |

Table 5: Zero-shot and few-shot Results.

| Extra PT Data | iVQA | MSRVTT-QA | MSVD-QA | TGIF-FrameQA |
|---|---|---|---|---|
| 250 | 7.2 | 5.5 | 18.7 | 26.7 |
| 2.5k | 20.4 | 15.8 | 30.3 | 28.4 |
| 25k | 28.1 | 16.2 | 31.6 | 31.0 |
| 250k | 31.8 | 18.1 | 32.3 | 44.1 |
| 2.5M | **35.7** | **20.3** | **38.7** | **55.6** |

Table 6: The influence of the pre-training dataset size. Zero-shot results are reported.

which consists of 21k noun phrases. For these noun phrases, we calculate the cosine similarity between each pair, and if the cosine similarity between two words exceeds 0.95, we remove the word with the smaller frequency. In addition, we manually removed less informative words from the vocabulary, such as "video," "picture," "photo," and so on.

## C  Results in Low-Data Regime

To examine the influence of the low-data regime on the performance of our method, we conducted experiments during both the pre-training and fine-tuning stages.

### C.1  Few-Shot Setting

We employed few-shot learning as an experimental setup to evaluate the performance of our method in the low-data regime during the fine-tuning stage. The experimental results under the few-shot setting are presented in Table 5. The experimental results reveal a substantial performance improvement when utilizing only 1% and 10% of the training data compared to the zero-shot setting.

### C.2  Size of the Pre-Training Dataset

We manipulated the size of the pre-training dataset and performed zero-shot testing to evaluate the performance of our method in the low-data regime during the fine-tuning stage. The detailed experimental results are presented in Table 6. The experimental results confirm the substantial impact of the pre-training dataset size on the model's performance. The performance of our method improves as the size of the pre-training dataset increases.

## D  The Influence of Misalignment

To examine the influence of modality misalignment, we conducted experiments that involved the separation of visual and textual modalities. The experimental results are presented in Table 7. In this experiment, we conducted pre-training using all modalities and evaluated the performance by separately considering the visual and textual modalities in the zero-shot setting.

From the perspective of the individual effects of the visual and textual modalities, the textual modality plays a more prominent role compared to the visual modality. This is primarily because we leverage the language model for multimodal fusion. In terms of question types, the textual modality demonstrates significant assistance, especially for questions of the "number", "where", and "when" types. These question types often require a higher level of abstraction, and the textual modality provides a more abstract representation compared to the visual modality. Conversely, the visual modality proves to be particularly advantageous for "who" type of questions, as this type of question requires the model to distinguish between visual entities present in the scene. The visual modality provides more detailed information about different visual entities, enabling the model to better address "who" type questions. Overall, the experimental results across all datasets consistently demonstrate that the synergistic alignment between the visual and textual modalities can effectively improve the performance.

## E  The Influence of Temporal Information

The temporal nature of videos is the fundamental distinction between videos and images. In this sec-

| | Benchmark | Modality | | Acc | What | Number | Color | Where | Who | When |
|---|---|---|---|---|---|---|---|---|---|---|
| | | Visual | Textual | | | | | | | |
| 1.1 | | ✓ | ✗ | 28.3 | — | — | — | — | — | — |
| 1.2 | iVQA | ✗ | ✓ | 34.3 | — | — | — | — | — | — |
| 1.3 | | ✓ | ✓ | **35.7** | — | — | — | — | — | — |
| 2.1 | | ✓ | ✗ | 18.3 | 11.6 | 20.9 | **12.9** | 9.4 | **35.4** | 3.6 |
| 2.2 | MSRVTT-QA | ✗ | ✓ | 19.4 | **13.1** | 65.1 | 11.9 | **12.6** | 31.8 | 4.6 |
| 2.3 | | ✓ | ✓ | **20.3** | **13.1** | **69.6** | 12.6 | 11.0 | 34.6 | **5.2** |
| 3.1 | | ✓ | ✗ | 34.8 | 25.7 | 29.5 | 50.0 | 46.4 | **51.7** | 1.7 |
| 3.2 | MSVD-QA | ✗ | ✓ | 37.1 | 30.2 | 62.6 | 37.5 | 50.0 | 47.6 | **10.3** |
| 3.3 | | ✓ | ✓ | **38.7** | **30.4** | **68.8** | 56.3 | **53.6** | 51.5 | 6.9 |
| 4.1 | | ✓ | ✗ | 31.9 | 42.3 | 4.5 | 41.4 | 22.4 | — | — |
| 4.2 | TGIF-FrameQA | ✗ | ✓ | 53.6 | 46.0 | 64.2 | 64.1 | 23.8 | — | — |
| 4.3 | | ✓ | ✓ | **55.6** | **46.6** | **69.8** | **65.5** | 23.8 | — | — |

Table 7: The influence of misalignment between visual and textual modalities. "Acc" refers to the overall Top-1 accuracy of the dataset.

| Sampling Frame Numbers | iVQA | MSRVTT-QA | MSVD-QA | TGIF-FrameQA |
|---|---|---|---|---|
| 1 | 26.2 | 16.0 | 34.2 | 48.4 |
| 2 | 31.3 | 19.4 | 37.4 | 51.4 |
| 4 | 33.6 | 19.8 | 37.6 | 53.4 |
| 8 | **35.7** | **20.3** | **38.7** | **55.6** |

Table 8: The infulence of the number of sampling frame numbers.

| | Benchmark | Temporal Prompt | Acc | What | Number | Color | Where | Who | When |
|---|---|---|---|---|---|---|---|---|---|
| 1.1 | iVQA | ✗ | 34.8 | — | — | — | — | — | — |
| 1.2 | | ✓ | **35.7** | — | — | — | — | — | — |
| 2.1 | MSRVTT-QA | ✗ | 19.3 | 12.9 | 59.4 | 12.3 | 10.2 | 32.7 | 4.5 |
| 2.2 | | ✓ | **20.3** | **13.1** | **69.6** | **12.6** | **11.0** | **34.6** | **5.2** |
| 3.1 | MSVD-QA | ✗ | 37.8 | 29.9 | 61.3 | **62.5** | 46.4 | 50.2 | **6.9** |
| 3.2 | | ✓ | **38.7** | **30.4** | **68.8** | 56.3 | **53.6** | **51.5** | **6.9** |
| 4.1 | TGIF-FrameQA | ✗ | 55.0 | **46.6** | 69.0 | 64.4 | 23.0 | — | — |
| 4.2 | | ✓ | **55.6** | **46.6** | **69.8** | **65.5** | **23.8** | — | — |

Table 9: Ablation studies on the expandable ordinal prompt approach.

tion, we examine the influence of temporal information in videos. Specifically, we will conduct experiments to investigate the influence of different sampling frame numbers and our proposed expandable ordinal prompt approach. When two frames are sampled, our method shows a notable improvement in performance as it becomes capable of capturing the temporal information present in the video. As the number of sampled frames increases, the performance of our method continues to improve, albeit with diminishing returns.

### E.1 The Influence of Sampling Frame Numbers

Table 8 presents the influence of sampling frame numbers. When only a single frame is sampled, the video question answering system degrades to an image question answering system, resulting in the poorest performance due to the inability to capture the temporal information of the video. As the number of sampled frames increases, the performance of our method continues to improve, albeit with diminishing returns. In conclusion, we have confirmed the importance of the temporal nature of videos.

### E.2 The Influence of the Expandable Ordinal Prompt Approach

Table 9 presents the influence of the expandable ordinal prompt approach. The experimental results demonstrate that the expandable ordinal prompt approach enhances the performance of nearly all question types across all datasets when compared to not utilizing the temporal prompt method. The experimental results confirm that our proposed expandable ordinal prompt approach facilitates the language model's acquisition of temporal information in videos.

| Ratio of Random Noise | 0% | 25% | 50% | 75% | 100% |
|---|---|---|---|---|---|
| Frame-Level Visual Features | 38.7 | 36.0 | 33.8 | 30.2 | 15.4 |
| Object-Level Visual Features | 38.7 | 38.2 | 38.1 | 38.0 | 37.4 |
| Frame-Level Textual Description | 38.7 | 38.6 | 38.0 | 37.2 | 35.4 |
| Object-Level Textual Description | 38.7 | 38.6 | 38.1 | 38.0 | 37.6 |

Table 10: Top-1 accuracy at different levels of random noise ratio in the MSVD-QA dataset.

| Ratio of Random Noise | 0% | 25% | 50% | 75% | 100% |
|---|---|---|---|---|---|
| Frame-Level Visual Features | 55.6 | 52.3 | 49.5 | 45.4 | 20.4 |
| Object-Level Visual Features | 55.6 | 54.7 | 53.6 | 51.2 | 48.4 |
| Frame-Level Textual Description | 55.6 | 55.5 | 54.5 | 52.8 | 48.5 |
| Object-Level Textual Description | 55.6 | 54.9 | 53.7 | 53.5 | 51.5 |

Table 11: Top-1 accuracy at different levels of random noise ratio in the TGIF-FrameQA dataset.

## F The Influence of Accumulated Errors

From a structural standpoint, our method conforms to a pipeline structure. At an equivalent scale, a pipeline structure, when compared to an end-to-end structure, offers increased flexibility. However, the pipeline structure method tends to result in a higher incidence of cascading errors.

To examine the influence of cascading errors, We designed experiments in which we introduced simulated errors by randomly substituting language information and adding random noise to visual information. Specifically, concerning the textual information of a sample, we randomly substitute sentences or words with sentences or words from another sample. Regarding the visual information of a sample, we randomly added Gaussian noise to the pixel images of frames or objects.

Table 1 presents the results on the MSVD-QA dataset, which emphasizes the overall video information. Table 2 presents the results on the TGIF-FrameQA dataset, which focuses on fine-grained details. Overall, the cascading errors generated by frame-level features surpass those of object-level features due to the dependence of object-level features on frame-level features. Similarly, the cascading errors arising from the visual modality outpace those from the textual modality, primarily attributed to the demanding visual capabilities required in video question answering tasks and the dependence of text description generation on visual features. Furthermore, as the ratio of random noise increases, the performance undergoes an accelerated decline rather than a linear descent. When the random noise ratio reaches 75%, our method still manages to maintain a relatively comparable performance. From a dataset perspective, the TGIF-FrameQA dataset exhibits heightened sensitivity to the cascading errors arising from object-level information in comparison to the MSVD-QA dataset. This differentiation stems from the fact that the TGIF-FrameQA dataset is designed to focus on intricate details, whereas the MSVD-QA dataset emphasizes the holistic video information.