# OpenReview forum: "Mulan: A Multi-Level Alignment Model for Video Question Answering"
_EMNLP/2023/Conference — EMNLP 2023 Findings_

### Official Review · Reviewer_eqr8 · 2023-08-01

**Typos Grammar Style And Presentation Improvements:** Line226
**Soundness:** 2

**Excitement:**

2: Mediocre: This paper makes marginal contributions (vs non-contemporaneous work), so I would rather not see it in the conference.

**Paper Topic And Main Contributions:**

This paper proposes multi-level alignment approach for VideoQA. It employs object mask generator and image captioning models to extract different levels of features. Extensive experiments are conducted to show the effectiveness of the proposed method.

**Questions For The Authors:**

- How is the static vocabulary V^s predefined? What is a set of candidate objects?
- Why should we distinguish static and dynamic vocabulary? What’s the functional difference between the two other than their naming?

**Reasons To Accept:**

- The proposed approach shows promising performance while using fewer parameters.
- Extensive experiments demonstrate the effectiveness of the proposed approach.

**Reasons To Reject:**

- Multi-level alignment doesn’t seem to be quite novel and previous works also have tried to use multi-level visual features (for example, [1]), not exactly same approach as the proposed one though. Therefore, a more extensive literature search would be needed to compare this work to previous studies. Employing similar concepts would be fine but the paper should show the difference to the prior works.

- There seems to be missing comparisons in Table 1. Although mPLUG-2 and the HiTeA variant with the best performance appear to perform better in the zero-shot setting, they are not included in the table. If its exclusion is due to its use of the VQA dataset, then a similar argument could be made about the proposed approach, as it also uses the MSCOCO, Visual Genome, Conceptual 12M, etc. dataset through its employment of the Cascade Mask R-CNN and BLIP. Hence, more thoughtfully designed experiments could have ensured a fairer comparison






[1] TVQA: Localized, Compositional Video Question Answering. Lei et al, 2018.

**Reproducibility:**

4: Could mostly reproduce the results, but there may be some variation because of sample variance or minor variations in their interpretation of the protocol or method.

**Reviewer Confidence:**

4: Quite sure. I tried to check the important points carefully. It's unlikely, though conceivable, that I missed something that should affect my ratings.

---

> ### Author Rebuttal · Authors · 2023-08-28
>
> We appreciate the reviewers' recognition of our experimental design and the positive experimental results.
>
> **Q1: The distinctions between our method and the method presented in [1].**
>
> Reviewers YRUS and 1gR7 found our method to be interesting. Particularly, reviewer yuTo explicitly acknowledged the novelty of our framework. The method presented in [1] is a multi-stream method. Despite both our proposed multi-level alignment method and this method utilizing multi-granularity features, there are significant differences in both motivation and specific methodologies. These distinctions are elaborated as follows:
>
> 1. Motivation: The method presented in [1] involves an initial encoding of various streams (such as regional visual features, ImageNet features, etc.) using LSTM, followed by the fusion of these streams. Consequently, the authors have described their methodology as a "multi-stream" method. Although features of varying granularity are extracted, features of the same granularity are merged into distinct streams before alignment. Due to the absence of alignment concept in multi-stream methods, the multi-stream approach significantly differs from the multi-level alignment concept in our method.
>
> 2. Specific Methodologies: The method presented in [1] employs an object detector to acquire region-based visual features and text information, whereas we propose a mask-guided visual feature encoding method to extract object-level visual features and text information. The motivation behind this design is to enable both frame-level and object-level features to be processed using a singular image encoder, thereby coexisting within the same semantic space. Moreover, this design encompasses an open vocabulary capability that is free from reliance on predefined object detection labels. It also enhances precision in object recognition, particularly for irregular objects.
>
> In the revised version, we will incorporate a comparison with multi-stream methods in the related work section.
>
> [1] TVQA: Localized, Compositional Video Question Answering. Lei et al, 2018.
>
> **Q2: Performance comparison with mPLUG-2 and the HiTeA variant in the zero-shot setting.**
>
> Both mPLUG-2 and HiTeA are indeed impressive works. Our perspective on the use of extra VQA dataset is as follows:
>
> 1. The intrinsic correlation between visual question answering tasks and video question answering highlights the significant benefit of incorporating extra VQA dataset. For instance, when comparing the HiTeA variant (with the VQA dataset) to HiTeA (without the VQA dataset), we observe substantial absolute performance improvements of 13.1% (from 8.6% to 21.7%) and 19.2% (from 18.2% to 37.4%) on the MSRVTT-QA and MSVD-QA datasets, respectively.
>
> 2. Following the reviewers' criteria, it can be stated that we make use of the MSCOCO and similar datasets through the integration of the Cascade Mask R-CNN and BLIP. However, it's worth noting that mPLUG-2 and HiTeA have also leveraged these datasets comprehensively.
>
> However, we concur that mPLUG-2 and the HiTeA variant represent impressive zero-shot video question answering methods. But a more comprehensive comparative analysis is essential. Hence, we have incorporated the results of these two methods into Table 1 of the revised version and provided elucidations in section 4.2. Moreover, we have supplemented a detailed analysis of dataset utilization in the appendix.
>
> **Q3: How is the static vocabulary V^s predefined? What is a set of candidate objects?**
>
> As mentioned in lines 908 to 917, we constructed the static vocabulary based on the class names from the OpenImage v7 dataset, which consists of 21k noun phrases. For these noun phrases, we calculate the cosine similarity between each pair, and if the cosine similarity between two phrases exceeds 0.95, we remove the phrase with the smaller frequency. In addition, we manually removed less informative phrases from the vocabulary, such as "video", "picture", "photo", and so on. "Candidate objects" refers to noun phrases within the static vocabulary.
>
> **Q4: Why should we distinguish static and dynamic vocabulary? What’s the functional difference between the two other than their naming?**
>
> As mentioned in lines 294 to 296, the predefined static vocabulary cannot cover all objects and attributes. In other words, employing a static vocabulary constitutes a closed vocabulary configuration, rendering it challenging to generate textual descriptions for uncommon objects. As a remedy, we extract noun phrases from global frame captions to form a dynamic vocabulary. Overall, in terms of functionality, the dynamic vocabulary serves to expand the generation of object descriptions into an open vocabulary, while the static vocabulary functions as a foundational resource when the quality of dynamic lexicon generation is compromised.
>
> **Q5:In line 226, $v^G$ shoule be $\phi_\text{CLIP}$?**
>
> We express our gratitude for the reviewer's examination. The phrasing at line 226 lacked clarity. We have rectified this in the revised version and conducted similar checks throughout the entire paper.

---

### Official Review · Reviewer_1gR7 · 2023-08-08

**Soundness:** 4

**Excitement:**

4: Strong: This paper deepens the understanding of some phenomenon or lowers the barriers to an existing research direction.

**Paper Topic And Main Contributions:**

This paper proposes a method to establish alignment between visual and textual modalities at different levels - object, frame, and video. At the object-level, the proposed method uses a mask-guided visual feature encoding method and a visual-guided text description method to learn fine-grained spatial information. At the frame-level, it combines visual features from individual frames with a caption generator to learn overall spatial information within the scene. At the video-level, an expandable ordinal prompt for textual descriptions is proposed, combined with visual features, to learn temporal information. Experimental results show that this method outperforms state-of-the-art methods, even with minimal visual language pre-training data and fewer trainable parameters.

**Reasons To Accept:**

I find the idea of utilizing different levels of video information to improve video question answering performance quite intriguing. This approach could enhance the model's understanding of spatial and temporal relations between objects and frames, as evidenced by promising experimental results.

**Reasons To Reject:**

I have some reservations about the suitability of this work for an EMNLP paper due to the following reasons:

(1) The authors only train the adapter's parameters, leaving the other model parameters untrained. It might be worth exploring why this decision was made and whether training all parameters could improve performance.
(2) Despite using strong pre-trained models like DeBERTa, the authors' model does not significantly outperform strong baselines such as mPLUG-2.
(3) Although the authors claim that their model can achieve better performance with less training data, it is worth noting that they only train the adapter's parameters. In contrast, more complex layers are trained in baselines like Flamingo and mPLUG-2.

**Reproducibility:**

2: Would be hard pressed to reproduce the results. The contribution depends on data that are simply not available outside the author's institution or consortium; not enough details are provided.

**Reviewer Confidence:**

4: Quite sure. I tried to check the important points carefully. It's unlikely, though conceivable, that I missed something that should affect my ratings.

---

> ### Author Rebuttal · Authors · 2023-08-28
>
> We appreciate the reviewers' recognition of the ideas, motivation, and experimental outcomes presented in our paper.
>
> **Q1: The authors only train the adapter's parameters, leaving the other model parameters untrained. It might be worth exploring why this decision was made and whether training all parameters could improve performance.**
>
> Utilizing only the adapter layer instead of full fine-tuning represents a parameter-efficient fine-tuning (PEFT) approach. The main purpose of Parameter-efficient Fine-tuning (PEFT) is to achieve a fine-tuning effect similar to that of tuning all parameters by adjusting only a small subset of parameters, aiming to reduce computational and storage costs. Experiments in [1] indicate that adjusting the adapter layer yields performance comparable to full fine-tuning. Moreover, parameter-efficient fine-tuning methods exemplified by Adapters are now widely employed.
>
> [1] Neil Houlsby, Andrei Giurgiu, Stanislaw Jastrzebski, Bruna Morrone, Quentin De Laroussilhe,
> Andrea Gesmundo, Mona Attariyan, and Sylvain Gelly. Parameter-efficient transfer learning
> for nlp. In ICML, 2019.
>
> **Q2: Although the authors claim that their model can achieve better performance with less training data, it is worth noting that they only train the adapter's parameters. In contrast, more complex layers are trained in baselines like Flamingo and mPLUG-2.**
>
> We concur with the reviewers' observations that, compared to other baselines, we exclusively fine-tuned the parameters of the adapter layer. The data concerning training parameters provides evidence for the potential of employing parameter-efficient fine-tuning techniques for adapting language models to multimodal fusion. Our method shows promise, as it can achieve state-of-the-art results even with a smaller dataset when compared to other baselines.
>
> **Q3: Despite using strong pre-trained models like DeBERTa, our model does not significantly outperform strong baselines such as mPLUG-2.**
>
> All the pre-trained models we employed lack the ability to address video-language tasks in isolation; for instance, DeBERTa operates solely as a monomodal language model. In the zero-shot setting, our method achieves an average accuracy that is 7.8% higher across all datasets compared to FrozenBiLM, despite both methods incorporating DeBERTa. This comparison with FrozenBiLM shows that in video question answering tasks, the capacity to understand visual-language interactions holds greater importance than linguistic comprehension alone.
>
> In our model, we have merely incorporated an additional 2.5 million video-language samples for video-language alignment. Compared to other strong baselines, we employed a minimal amount of video-language datasets. Nevertheless, our model attains state-of-the-art performance. Considering the dataset size, this outcome is profoundly encouraging.

---

### Official Review · Reviewer_YRUS · 2023-08-11

**Soundness:** 3

**Excitement:**

3: Ambivalent: It has merits (e.g., it reports state-of-the-art results, the idea is nice), but there are key weaknesses (e.g., it describes incremental work), and it can significantly benefit from another round of revision. However, I won't object to accepting it if my co-reviewers champion it.

**Paper Topic And Main Contributions:**

This paper focuses on the Video Question Answering task. Different from previous methods that target to improve joint representations of video and text, this paper establishs alignment between visual and textual modalities at multiple visual levels.

**Reasons To Accept:**

The idea is interesting.
Results shown in this paper are promising.

**Reasons To Reject:**

My concerns are as follows:
1) why there is no loss function in this paper?
2) in page 12, how to use the learning rate, epoch, batch size, Adam optimizer without loss function?
3) in line 382, authors state that they update the parameters of the visual-to-text projection modules P and the adapter module. Could you provide more details abou how to update the parameters?
4) in the zero-shot setting, how to conduct the video-level alignment? If multiple videos are relevant to the same text description or multiple text descriptions are relevant to the same video, how to align videos and text descriptions?

**Reproducibility:**

2: Would be hard pressed to reproduce the results. The contribution depends on data that are simply not available outside the author's institution or consortium; not enough details are provided.

**Reviewer Confidence:**

4: Quite sure. I tried to check the important points carefully. It's unlikely, though conceivable, that I missed something that should affect my ratings.

---

> ### Author Rebuttal · Authors · 2023-08-28
>
> We extend our gratitude to the reviewer for acknowledging the merit of our proposed idea and recognizing the promising experimental results.
>
> **Q1: The loss function.**
>
> As mentioned in lines 383 to 389, we employ a masked language modeling objective as our loss function, following the setup established in BERT. The mathematical formulation of the masked language modeling will be provided in the revised version.
>
> **Q2: The update process of the visual-to-text projection module $P$.**
>
> As mentioned in lines 369 to 375, the projection matrices $P$ ($P^G$ and $P^L$) correspond to a linear layer, and lines 373 to 375 describe their update process.
>
> **Q3: The update process of the adapter module.**
>
> As mentioned in lines 364 to 365, we use the same Adapter layer as in the FrozenBiLM. In specific terms, for the hidden state $z$, the update process of $Adapter(z)$ is as follows:
>
> $Adapter(z) = z + W^{up}\psi(W^{down}z)$
>
> In this process, $W^{down} \in \mathbb{R}^{D\times D_h}$, $W^{up} \in \mathbb{R}^{D_h\times D}$, $D$ is the hidden dimension of the transformer, $D_h$ is the bottleneck dimension, and $\psi$ is a ReLU activation function.
> $D_h$ is typically set to be smaller than $D$ such that the adapters are lightweight.
>
> **Q4: In the zero-shot setting, how to conduct the video-level alignment?**
>
> As mentioned in Figure 1, during the training phase, for a video-text sample, video-level alignment refers to aligning all sampled frames with the provided original textual description within the sample. However, during the evaluation phase, where the model receives a question and a video segment as input, video-level alignment is not applicable. A comprehensive explanation of video-level alignment will be incorporated in the revised version.
>
> **Q5: If multiple videos are relevant to the same text description or multiple text descriptions are relevant to the same video, how to align videos and text descriptions?**
>
> In video-language tasks, a standard sample generally involves a single video paired with a corresponding textual description. Hence, our model's input consists of an individual video alongside its associated text. If there are multiple text descriptions relevant to the same video, each description is treated as an independent sample. This principle is similarly applied in cases where multiple videos are related to the same text description. This processing approach is widely employed in the field of video question answering.
>
> Considering all samples within the dataset, we concur with the scenarios highlighted by the reviewers. Instances where multiple text descriptions are relevant to the same video augment the diversity of video-text descriptions. This increase in textual descriptions assists in alleviating the challenge of information imbalance between modalities, thus contributing to enhanced performance. In cases where multiple videos are related to the same text description, Due to video content varies, textual descriptions at the frame and object levels also differ. This disparity enables our method to effectively differentiate between multiple videos sharing the same textual description.

---

### Official Review · Reviewer_yuTo · 2023-08-11

**Soundness:** 4

**Excitement:**

3: Ambivalent: It has merits (e.g., it reports state-of-the-art results, the idea is nice), but there are key weaknesses (e.g., it describes incremental work), and it can significantly benefit from another round of revision. However, I won't object to accepting it if my co-reviewers champion it.

**Paper Topic And Main Contributions:**

This paper is about Video Question Answering. The author introduces a multi-level visual-language alignment method for video question answering, which provides a novel framework to learn spatial information and contains object-level, frame-level and video-level character.

**Questions For The Authors:**

The method is relatively complex, and I have the following questions:
1)The model is relatively complex, and under the same settings, is it comparable in terms of running time with previous methods？
2) I think this is a pipeline structure, whether the accumulation of errors can be evaluated, and which part of it most affects the results
3) Will the method in 3.3 lead to excessive dependence of object level features on frame level features?

**Reasons To Accept:**

This paper propose a method Mulan: a Multi-Level Alignment Model for Video Question Answering, which establishes alignment between visual and textual modalities at the object-level, frame-level, and video-level.  The experiment is sufficient and experimental results illustrate that this method surpasses state-of-the-art baselines.
The framework proposed in this paper is novel, which Incorporates rich features to extract information in video.

**Reasons To Reject:**

1.The article is well organized overall, but there are unclear parts, such as the third paragraph of the introduction.
2.The innovation of the article is mainly reflected in ‘first work in the field of video question answering that explores multi-level visual-language alignment’, but the specific method innovation is average, or it is the previous work.


**Reproducibility:**

3: Could reproduce the results with some difficulty. The settings of parameters are underspecified or subjectively determined; the training/evaluation data are not widely available.

**Reviewer Confidence:**

4: Quite sure. I tried to check the important points carefully. It's unlikely, though conceivable, that I missed something that should affect my ratings.

---

> ### Author Rebuttal · Authors · 2023-08-28
>
> We express our appreciation to the reviewer for the valuable experimental suggestions as well as the recognition of the novelty of our framework.
>
> **Q1: The innovation of the article is mainly reflected in ‘first work in the field of video question answering that explores multi-level visual-language alignment’, but the specific method innovation is average, or it is the previous work.**
>
> As highlighted by the reviewer, one of our major contributions is the novelty of the framework. Another important novelty of our method primarily lies in the proposed mask-guided visual feature encoding method for object-level feature extraction, as opposed to the direct use of object detectors. The motivation behind this design is to enable both frame-level and object-level features to be processed using a singular image encoder, thereby coexisting within the same semantic space. Moreover, this design encompasses an open vocabulary capability that is free from reliance on predefined object detection labels. It also enhances precision in object recognition, particularly for irregular objects.
>
> **Q2: The article is well organized overall, but there are unclear parts, such as the third paragraph of the introduction.**
>
> We have already revised the third paragraph of the introduction, which will be reflected in the revised version. Specifically, we relocated certain minor detailed descriptions from the third paragraph of the introduction to Section 3.1 and provided additional explanations. Furthermore, the three authors of this paper have collectively reviewed the entire paper and made corrections, encompassing sections beyond the third paragraph of the introduction.
>
> **Q3: The model is relatively complex, and under the same settings, is it comparable in terms of running time with previous methods?**
>
> In our experimental environment and settings, the average inference time for a single sample in the MSRVTT-QA dataset is 780 ms. For comparison, the average inference time for FrozenBiLM, which has fewer parameters, is 417 ms. For larger-scale models, it is typically observed that longer inference times are incurred.
>
> **Q4: Our method is a pipeline structure, whether the accumulation of errors can be evaluated, and which part of it most affects the results.**
>
> We concur that our method adheres to a pipeline structure. At an equivalent scale, a pipeline structure, in comparison to an end-to-end structure, tends to incur more cascading errors. However, the pipeline structure approach affords greater flexibility.
>
> Simultaneously, we express our gratitude for the reviewer's supplementary suggestions regarding auxiliary experiments. We designed experiments in which we introduced simulated errors by randomly adding noise to visual information and substituting language information. This allowed us to conduct a cascade error analysis. The experimental outcomes elucidate that the cascading errors engendered by frame-level features exceed those originating from object-level features. Similarly, the cascading errors arising from the visual modality outpace those from the textual modality. Among them, the most pronounced cascading errors are induced by the frame-level visual features. We will integrate the aforementioned experiments and furnish detailed experimental data, along with more comprehensive experimental setups and analyses, in the appendix of the revised version.
>
> **Q5: Will the method in 3.3 lead to excessive dependence of object level features on frame level features?**
>
> Section 3.3 delineates the process of text description generation. The dependency process in text description generation can be described as follows: object-level text descriptions depending on both the dynamic and static vocabularies, with the dynamic vocabulary depending on frame-level text descriptions. Hence, we posit that object-level text descriptions are dependent on frame-level text descriptions. However, due to the presence of a predefined static vocabulary, the model retains a certain degree of text generation capability even when the quality of the dynamic vocabulary is subpar. Consequently, this dependency is not excessive.

---

### Meta-Review · Area_Chair_ADJL · 2023-09-18

**Recommendation:** 3

**Metareview:**

The manuscript introduces a new approach to video QA by aligning visual and textual at different levels to accomplish the task. Reviewers and authors engaged in a productive discussion around strength and weaknesses of the approach and some details of the claims. The idea of aligning textual and visual information is novel and most important aspect of the manuscript. Some of the baselines highlighted by the reviewers were not in the original submission and but authors have produced comparison and used the arguments in responses to the reviewers. While this helped with the discussion, a more comprehensive literature survey would have been better in the submission. One of the main advantages of the proposed approach is the claim of using small pre-training data compared with the state of the art to achieve the same performance. However, this claim needs to be adjusted to refer only to the extra pre-training data. Technically, encoders are pre-trained with additional data before being used.
In summary authors have identified a better approach to leverage the information in a video to answer question. However while we see some improvements over state of the art, this is not consistent potentially highlighting the need for further experimentation to improve the approach.

---

### Decision · Program_Chairs · 2023-10-07

**Decision:**

Accept-Findings

**Comment:**

The manuscript introduces a new approach to video QA by aligning visual and textual at different levels to accomplish the task. Reviewers and authors engaged in a productive discussion around strength and weaknesses of the approach and some details of the claims. The idea of aligning textual and visual information is novel and most important aspect of the manuscript. Some of the baselines highlighted by the reviewers were not in the original submission and but authors have produced comparison and used the arguments in responses to the reviewers. While this helped with the discussion, a more comprehensive literature survey would have been better in the submission. One of the main advantages of the proposed approach is the claim of using small pre-training data compared with the state of the art to achieve the same performance. However, this claim needs to be adjusted to refer only to the extra pre-training data. Technically, encoders are pre-trained with additional data before being used.
In summary authors have identified a better approach to leverage the information in a video to answer question. However while we see some improvements over state of the art, this is not consistent potentially highlighting the need for further experimentation to improve the approach.